# Salivary Diagnosis of Dental Caries: A Systematic Review

**Rita Antonelli** [1,*] , **Valentina Massei** [1], **Elena Ferrari** [2], **Mariana Gallo** [2] , **Thelma A. Pertinhez** [2,*] , **Paolo Vescovi** [1], **Silvia Pizzi** [1] and **Marco Meleti** [1]

1    Centro Universitario di Odontoiatria, Department of Medicine and Surgery, University of Parma, 43126 Parma, Italy; valentina.massei@studenti.unipr.it (V.M.); paolo.vescovi@unipr.it (P.V.); s.pizzi@unipr.it (S.P.); marco.meleti@unipr.it (M.M.)

2    Laboratory of Biochemistry and Metabolomics, Department of Medicine and Surgery, University of Parma, 43126 Parma, Italy; elena.ferrari@unipr.it (E.F.); mariana.gallo@unipr.it (M.G.)

\*    Correspondence: rita.antonelli@unipr.it (R.A.); thelma.deaguiarpertinhez@unipr.it (T.A.P.); Tel.: +39-3469692224 (R.A.)

**Abstract:** The activity of dental caries, combined with its multifactorial etiology, alters salivary molecule composition. The present systematic review was developed to answer the following question: "Are salivary biomarkers reliable for diagnosis of dental caries?". Following the "Preferred Reporting Item for Systematic Reviews and Meta-analysis" (PRISMA) guidelines, the review was conducted using multiple database research (Medline, Web of Science, and Scopus). Studies performed on healthy subjects with and without dental caries and providing detailed information concerning the clinical diagnosis of caries (Decayed, Missing, Filled Teeth-DMFT and International Caries Detection and Assessment System-ICDAS criteria) were included. The quality assessment was performed following a modified version of the Joanna Briggs Institute Prevalence Critical Appraisal Checklist. The protocol was registered in the International Prospective Register of Systematic Reviews (PROSPERO, ID: CRD42022304505). Sixteen papers were included in the review. All studies reported statistically significant differences in the concentration of salivary molecules between subjects with and without caries ($p < 0.05$). Proteins were the most investigated molecules, in particular alpha-amylase and mucins. Some studies present a risk of bias, such as identifying confounding factors and clearly defining the source population. Nevertheless, the 16 papers were judged to be of moderate to high quality. There is evidence that some salivary compounds studied in this review could play an important diagnostic role for dental caries, such as salivary mucins, glycoproteins (sCD14), interleukins (IL-2RA, 4,-13), urease, carbonic anhydrase VI, and urea.

**Keywords:** biomarkers; caries detection; cariology; dental caries; salivary diagnostics





## 1. Introduction

Dental caries (also known as "tooth decay") is a pathological process consisting of the demineralization of dental hard tissues and the formation of a dental cavity, often causing pain and, if not treated, tooth loss [1].

Even if its incidence in the population between 5 and 12 years and 25 and 44 years in Western countries has decreased in the last four decades [2], dental caries remains one of the most widespread infectious diseases. It has been estimated that there are approximately 2.4 billion people with untreated cavities in permanent teeth, as well as 621 million children with caries in deciduous teeth [3].

Dental caries has a multifactorial etiology. Two primary causal factors are the frequent consumption of free sugars and the metabolism of some commensal tooth-adherent bacteria. Several oral microorganisms, by metabolizing fermentable carbohydrates, produce organic acids. Such acids cause a decrease in salivary pH, inducing tooth tissue demineralization, starting from the external surface of the enamel. A remarkable variety of microorganism species associated with dental caries has been identified, including *Streptococcus mutans*,

which is the most prevalent bacteria found in subjects with dental caries (58.3%) [4]. However, many other host and behavioral factors, such as enamel defects, alterations of salivary pH, flow rate and composition, poor oral hygiene, and low socioeconomic level also play an important role in dental caries pathogenesis [1].

An accurate diagnosis, screening programs, and individual risk assessments are fundamental to preventing this pathology and reducing the number of untreated caries. Currently, the diagnosis of dental caries is mainly based on clinical inspection and, when necessary, on radiographic examination.

The World Health Organization (WHO) recommended using the decayed, missing, and filled teeth (DMFT) index to monitor the distribution and prevalent trends of dental caries. Such an index is based on clinical examination and reflects a patient risk profile for developing dental caries [5].

The International Caries Detection and Assessment System (ICDAS) was developed in 2005 to integrate different systems into a single index that provides information on dental caries stage, activity, and risk profile. The system comprises seven codes, ranging from code 0 (healthy teeth) to codes 5 and 6, indicating caries with exposed dentine [6]. The ICDAS also requires a specific tool (laser-induced fluorescence—Diagnodent™ Pen) to assess tooth decay's stage and gravity.

In the last 20 years, the interest in new non-invasive techniques for the detection of caries, like quantitative light fluorescence (QLF), digital fiber-optic transillumination (DIFOTI—KaVo Diagnocam™), and electric conductance (EC), has significantly increased [7]. At the same time, the interest in the diagnostic role of saliva has significantly increased.

Saliva is a complex biological fluid produced by parotid (20%), submandibular (65–70%), sublingual (7% to 8%), and minor salivary glands located in the lips, tongue, palate, cheeks, and pharynx (<10%), as well as by gingival sulcus (crevicular fluid) [8]. Saliva secretion, its flow rate, and composition depend on several factors, including the type and size of glands, nutritional status, gender, age, and emotional state. Whole saliva (WS) consists of a pool of different organic and inorganic components which represent the physiology of the human body [9]. Besides the massive presence of water, saliva contains metabolites, enzymes, antibodies, hormones, antimicrobial molecules, and cytokines, all potential biomarkers for oral and systemic diseases [10].

Biomarkers include molecules belonging to the genome, epigenome, transcriptome, proteome, and metabolome that may be useful for monitoring health status, disease diagnosis and prognosis, or evaluating response to treatment [11]. Emerging evidence suggests that saliva is an excellent and innovative matrix for the search of molecular markers due to its intrinsic properties (e.g., the complexity of composition, abundance of molecules, fluid availability), combined with its non-invasive sampling and ease of collection, transport, and storage [8].

The present systematic review aims to answer the question: "Are salivary biomarkers reliable for the diagnosis of dental caries?" formulated according to the "Population or problem", "Intervention or exposure", "Comparison", "Outcome" (PICO) worksheet.

## 2. Materials and Methods

The Preferred Reporting Item for Systematic Reviews and Meta-Analysis (PRISMA) guidelines was used to conduct the present review [12]. The protocol was registered in the International Prospective Register of Systematic Reviews (PROSPERO, Centre for Reviews and Dissemination, York, UK, ID: CRD42022304505).

### 2.1. Search Strategy

We searched Medline, Scopus, and Web of Science databases for original scientific papers published in English after 2000. The search terms were "saliva" or "salivary biomarkers", combined through the Boolean indicator "AND" with "tooth decay" and "dental caries". Periodic screenings of the databases were performed between August 2021 and Oc-

tober 2023. Duplicates were eliminated using End Note X9© software (Clarivate Analytics, Philadelphia, USA, London, UK).

Two independent researchers performed a first-level screening by evaluating titles and abstracts. Conference proceedings, meeting abstracts, short communication, editorials, letters to the editor, and reviews were excluded. Only studies performed on humans, providing detailed information on the clinical diagnosis of dental caries, and specifically applying DMFT and/or ICDAS criteria were included. References in literature reviews were also screened to identify other possible papers of interest. Final eligibility was assessed through full-text evaluation according to the exclusion and inclusion criteria summarized in Table 1.

**Table 1.** Inclusion and exclusion criteria.

| Inclusion Criteria | Exclusion Criteria |
| --- | --- |
| English language | Systematic reviews, letters to the editor, editorials, short communications, meeting abstracts |
| Papers published after June 2000 | Studies on pH and buffering capacity evaluation |
| Studies on humans | Studies on the inorganic composition of saliva (ions) |
| | Studies on "risk of caries" |
| Studies applying DMFT and/or ICDAS criteria | Studies on the evaluation of deciduous teeth and/or early childhood caries (ECC) |
| | Studies on the assessment of the caries level of the child in relation to the mother |
| | Studies on microbiome and bacteria |

Studies focused on the pH, buffering capacity, and inorganic composition of saliva were excluded. Papers evaluating the risk of caries, deciduous teeth, early childhood caries (ECC), and the caries level of the child in relation to the caries experience of the mother were excluded. Additionally, we excluded studies focused on dental plaque and oral microbiome and studies performed on subjects with systemic disorders and/or oral diseases other than caries and on alcohol and/or drug users. Papers on medical devices (for experimental purposes) or involving the administration of products for therapeutic uses were also excluded.

Selected studies underwent data extraction and critical appraisal.

*2.2. Data Extraction*

Data extracted from each study were summarized into Excel®tables.

Table 2 shows the general characteristics of the selected studies, including the title, authors, year of publication, type of study, and main clinical features of the evaluated subjects.

**Table 2.** General characteristics of selected studies.

| Authors | Title | Design of Study | N° Subjects |
| --- | --- | --- | --- |
| Ahmadi-Motamayel et al., 2018 [13] | Salivary and Serum Antioxidant and Oxidative Stress Markers in Dental Caries | Case–control study | 56 CG (M:F = 28:28; 17 y)<br>- DMFT = 0<br>62 AC (M:F = 27:35; 17 y)<br>- DMFT ≥ 5 |
| Ashwini et al., 2020 [14] | Dentin degradonomics—The potential role of salivary MMP-8 in dentin caries | Case–control study | 25 CG<br>- DMFT = 0<br>50 AC<br>- 25 (caries not involving more than two teeth)<br>- 25 (caries involving more than three teeth) |
| Ayad et al., 2000 [15] | The Association of Basic Proline-Rich Peptides from Human Parotid Gland Secretions with Caries Experience | Case–control study | 9 CG (M:F = 4:5; 59.2 y)<br>- DMFT = 0<br>9 AC (M:F = 4:5; 51.2 y)<br>- DMFT = 38.4 |

**Table 2.** *Cont.*

| Authors | Title | Design of Study | N° Subjects |
|---|---|---|---|
| Banderas-Tarabay et al., 2002 [16] | Electrophoretic Analysis of Whole Saliva and Prevalence of Dental Caries. A Study in Mexican Dental Students | Case–control study | 24 CG (19 y)<br>- DMFT < 4<br>40 AC (19 y)<br>- DMFT > 10 |
| Bilbilova et al., 2012 [17] | Correlation between Salivary Urea Level and Dental Caries | Case–control study | 40 CG (16 y)<br>- DMFT = 0–3<br>40 AC (16 y)<br>- DMFT > 10 |
| Gabryel-Porowska et al., 2014 [18] | Mucin Levels in Saliva of Adolescents with dental caries | Case–control study | 8 CG (18 y)<br>- DMFT = 3<br>27 AC (18 y)<br>- DMFT > 11 |
| Gornowicz et al., 2012 [19] | Pro-Inflammatory Cytokines in Saliva of Adolescents with Dental Caries Disease | Case–control study | 10 CG (18 y)<br>- Dmft = 0<br>27 AC (18 y)<br>- DMFT = 11.33 |
| Gornowicz et al., 2014 [20] | The Assessment of sIgA, Histatin-5, and Lactoperoxidase Levels in Saliva of Adolescents with Dental Caries | Case–control study | 8 CG (18 y)<br>- DMFT = 3<br>27 AC (18 y)<br>- DMFT > 11 |
| Kulhavá et al., 2020 [21] | Proteomic Analysis of Whole Saliva in Relation to Dental Caries Resistance | Case–control study | 12 CG ($31.8 \pm 7.6$ y)<br>- DMFT = 0–1<br>15 AC ($38.4 \pm 5.6$ y)<br>- DMFT = 7–12 |
| Mira et al., 2017 [22] | Salivary Immune and Metabolic Marker Analysis (SIMMA): A Diagnostic Test to Predict Caries Risk | Case–control study | 10 CG (19–39 y)<br>10 AC (19–39 y) |
| Nireeksha et al., 2017 [23] | Salivary Proteins as Biomarkers in Dental Caries: In Vivo study | Case–control study | 20 CG (25–40 y)<br>60 AC (25–40 y)<br>- 20 DMFT = 1–3<br>- 20 DMFT = 4–10<br>- 20 DMFT > 10 |
| Paqué et al., 2021 [24,25] | Salivary Biomarkers for Dental Caries Detection and Personalized Monitoring | Case–control study | 18 CG<br>38 AC |
| Piekoszewska-Ziertek et al., 2020 [25] | Polymorphism in the CAVI Gene, Salivary Properties and Dental Caries | Case–control study | 9 CG ($13.25 \pm 1.72$ y)<br>- DMFT = 0<br>121 AC ($13.25 \pm 1.72$ y)<br>- DMFT > 0 |
| Prester et al., 2017 [26] | Salivary sCD14 as a Potential Biomarker of Dental Caries Activity in Adults | Case–control study | 25 CG (35 y)<br>- DMFT = 16.5<br>- Cavities = 6.8<br>30 AC (31 y)<br>- DMFT = 13.8<br>- Cavities = 0 |
| Reyes et al., 2014 [27] | Caries-Free Subjects Have High Levels of Urease and Arginine Deiminase Activity | Cross-sectional study | 10 CG<br>- DMFT = 0<br>12 AC<br>- DMFT $\geq$ 4 |
| Yazid et al., 2020 [28] | Caries Detection Analysis in Human Saliva Alpha Amylase | Case–control study | 12 CG (18–55 y)<br>- ICDAS = 0<br>15 AC (18–55 y)<br>- ICDAS $\geq$ 4 |

Legend: CG = control group; AC = active caries; F = females; M = males; y = years old; DMFT = decayed, missing, and filled tooth index, ICDAS = International Caries Detection and Assessment System score.

Table 3 displays the type of saliva sample, the conditions for saliva collection, and the analytical procedures used for identifying and quantifying the salivary compounds (e.g., the type of specimen, method of collection, and analysis).

**Table 3.** Analysis of saliva sample, conditions for saliva collection, and the analytical procedures used.

| Authors | Typology of Saliva Sample | Saliva Collection | Biomarker of Analysis |
|---|---|---|---|
| Ahmadi-Motamayel et al., 2018 [13] | Unstimulated whole saliva | Samples were obtained by spitting for 5 min. | Sialo-chemical analysis |
| Ashwini et al., 2020 [14] | Stimulated saliva | Samples were collected after chewing on a paraffin wax for 5 min. | ELISA |
| Ayad et al., 2000 [15] | Stimulated parotid (ductal) saliva | Samples were collected in the morning. Subjects were requested not to eat for 2 h before collection. Gustatory stimulated secretions were obtained by means of sugar-free lemon drops. | HPLC |
| Banderas-Tarabay et al., 2002 [16] | Unstimulated whole saliva | Samples were collected in the morning. Subjects refrained from eating, drinking, smoking, and oral hygiene for at least 2 h prior to saliva collection. | Electrophoresis |
| Bilbilova et al., 2012 [17] | Unstimulated whole saliva Food-stimulated whole saliva | Samples were collected in the morning, in the fasted state, and without oral hygiene. Samples were taken from all the participants at different time intervals: 5, 30, and 60 min after the meal. | Urase-based enzymatic method |
| Gabryel-Porowska et al., 2014 [18] | Unstimulated whole saliva | Samples were collected in the morning. Subjects abstained from eating and drinking for 2 h. Unstimulated whole saliva was collected for 10 min by a spitting method. | ELISA |
| Gornowicz et al., 2012 [19] | Unstimulated whole saliva | Samples were collected in the morning. Subjects abstained from eating and drinking for 2 h. Samples were collected for 10 min by a spitting method. | ELISA |
| Gornowicz et al., 2014 [20] | Unstimulated whole saliva | Samples were collected in the morning. Subjects abstained from eating and drinking for 2 h. Samples were collected by a standard method in sterilized tubes (placed on ice after collection). | ELISA |
| Kulhavá et al., 2020 [21] | Unstimulated whole saliva | Samples were collected in the morning. Volunteers were requested not to eat or drink and brush their teeth for 1–2.5 h prior to the trial. | LC-MS |
| Mira et al., 2017 [22] | Unstimulated whole saliva | Five milliliters of non-stimulated saliva samples were taken by drooling at 30 min, 6, 12, and 24 h after toothbrushing in a sterile 50 mL tube (avoiding spitting or plaque removal with the tongue). | ELISA |
| Nireeksha et al., 2017 [23] | Unstimulated whole saliva | Samples were collected in the morning. Subjects were asked to abstain from toothbrushing, using mouthwash, and eating/drinking for 2 h prior to sample collection. | PAGE |
| Paqué et al., 2021 [24,25] | Unstimulated whole saliva | Samples were collected in the morning. The participants were asked not to eat, drink sugary drinks, or perform any oral hygiene measures the night before the saliva donation. Water intake was permitted. | ELISA |
| Piekoszewska-Ziertek et al., 2020 [25] | Unstimulated whole saliva Buccal smear | Samples were collected in the morning. Subjects were instructed to fast for at least 2 h and not to use antibacterial mouth rinse. The smear was collected for about 2 min using a special buccal swab. | ELISA RT-PCR |
| Prester et al., 2017 [26] | Unstimulated whole saliva (n = 55) Stimulated whole saliva (n = 55) | Samples were collected between 8–11 a.m. and 2–4 p.m. Two hours before collection, the participants were asked to refrain from eating, drinking, smoking, and toothbrushing to obtain a relatively constant baseline. The resting saliva was collected before chewing 5 g of pure paraffin wax for five minutes without swallowing. | ELISA |

**Table 3.** *Cont.*

| Authors | Typology of Saliva Sample | Saliva Collection | Biomarker of Analysis |
|---|---|---|---|
| Reyes et al., 2014 [27] | Unstimulated whole saliva | Saliva sample was collected by expectorating 3 mL of saliva in a sterile plastic tube. Subjects were instructed not to eat for 12 h prior to sample collection and to abstain from any type of oral hygiene. | Biochemical analysis and spectrophotometry (Thermo Spectronic Unicam UV-530 UV–visible) |
| Yazid et al., 2020 [28] | Unstimulated whole saliva | Subjects were instructed to accumulate saliva and drop it into a cryovial (about 2 mL). A protease inhibitor was added to the saliva samples. | UV-Vis spectroscopy |

Legend: ELISA = enzyme-linked immunosorbent assay; HPLC = high-performance liquid chromatography; RT-qPCR = real-time polymerase chain reaction; LC-MS = liquid chromatography–mass spectrometry; PAGE = polyacrylamide gel electrophoresis.

Table 4 shows the statistical association between specific salivary molecules and dental caries.

**Table 4.** Statistical association between specific salivary molecules and dental caries.

| Authors | Biomarker Category | Biomarker | Statistical Association with Dental Caries (P) |
|---|---|---|---|
| Ahmadi-Motamayel et al., 2018 [13] | Metabolite | MDA | 0.001a (higher in AC) |
| Ashwini et al., 2020 [14] | Protein | MMP8 | < 0.05a (higher in AC) |
| Ayad et al., 2000 [15] | Protein | Ps1 (PRB1) Con1 (PRB2) Pmo1 (unassigned gene) | Ps1 = < 009a (higher in CF) Con1 < 009a (higher in CF) Pmo1 < 015a (higher in CS) |
| Banderas-Tarabay et al., 2002 [16] | Protein | MG1 MG2 PRP 1 | < 0.001 (lower in CS) |
| Bilbilova et al., 2012 [17] | Metabolite | Urea | < 0.01a (higher in CF) |
| Gabryel-Porowska et al., 2014 [18] | Protein | MUC1 MUC5B MUC7 | MUC1 = 0.011 a (higher in AC) MUC5B = 0.06 (higher in AC) MUC7 =0.918 (higher in CG) |
| Gornowicz et al., 2012 [19] | Protein | IL-6 IL-8 TNF-α | IL-6 < 0.005 a (higher in AC) IL-8 < 0.008 a (higher in AC) TNF-α < 0.002 a (higher in AC) |
| Gornowicz et al., 2014 [20] | Protein | SIgA Histatin-5 LPO | SIgA = 0.003 a (higher in CA) Histatin-5 = 0.015 a (higher in CA) LPO = 0.02 a (higher in CA) |
| Kulhavá et al., 2020 [21] | Protein | Proteins in salivary supernatants: α-Amylase 1 Serum albumin Protein S100-A9 Immunoglobulin heavy variable 4–31 Immunoglobulin heavy constant α 1 Immunoglobulin κ constant Apolipoprotein A-I Immunoglobulin heavy variable 1–44 Cystatin B Lysozyme C Annexin A1 Polymeric immunoglobulin receptor Prolactin-inducible protein Proteins in salivary pellets: Annexin A1 Protein ς Cornulin | P not reported a (higher in CF) P not reported a (higher in AC) |

**Table 4.** *Cont.*

| Authors | Biomarker Category | Biomarker | Statistical Association with Dental Caries (P) |
|---|---|---|---|
| Mira et al., 2017 [22] | Protein Metabolite | LL37 IgA Statherin β-defensin 2 Collagen I Fibrinectin Formate | < 0.5 [a] |
| Nireeksha et al., 2017 [23] | Protein | Total protein IgA Mucin CRP Albumin globulin | Total protein < 0.001 [a] (higher in CF) IgA < 0.001) (higher in CF) Mucine < 0.01 [a*] (higher in AC) CRP < 0.01 [a**] (higher in AC) Albumin globulin < 0.001 [a***] (higher in AC) |
| Paqué et al., 2021 [24,25] | Protein | IL-4 IL-13 IL-2-RA Eotaxin (CCL11) | IL-4 = $4.1 \times 10^{-13a}$ IL-13 = $3.1 \times 10^{-12a}$ IL2-RA = $1.0 \times 10^{-4a}$ Eotaxin (CCL11) = $4.4 \times 10^{-4a}$ |
| Piekoszewska-Ziertek et al., 2020 [25] | Protein / Gene | CA VI rs2274333 A/G | CA VI = 0.014 [a] (lower in AC) rs2274333 A/G < 0.5 [a] (higher in AC) |
| Prester et al., 2017 [26] | Protein | sCD14 | 0.004 [a] in resting saliva (higher in AC) 0.001 [a] in stimulated saliva (higher in AC) |
| Reyes et al., 2014 [27] | Protein | Urease activity ADS activity | Urease activity 0.01 [a] (higher in CF) ADS activity = 0.02 [a] - Higher in CF |
| Yazid et al., 2020 [28] | Protein | Alpha-amylase | P not reported [a] (higher in AC) |

Legend: AC = active caries; ADS = urease and arginine deiminase system; CA VI = carbonic anhydrase isozyme VI; CF = caries-free; CG = control group; CRP = C-reactive protein; CS = caries-susceptible; LPO = human lactoperoxidase; MDA = malondialdehyde; MG1 = high-molecular-weight mucin; MG2 = low-molecular-weight mucin; MUC1 = mucin-1; PRP-1 = acidic proline-rich protein-1; sCD14 = soluble form of CD14 (coreceptor); SI = statistically insignificant; SIgA = salivary IgA; SS = statistically significant. [a] Statistically significant. * Except between CF group and AC group I (DMFT = 1–3). ** Except for CF group and CA group I–group II (DMFT 1–10). *** Except for AC group II and AC group III (DMFT > 4).

### 2.3. Critical Appraisal

For critical appraisal, we used two modified versions of the Joanna Briggs Institute Prevalence Critical Appraisal Checklist (JBI critical appraisal) adapted to the scopes of the present review, one for case–control studies and another for analytical cross-sectional studies [29]. The detailed description of all questions from the JBI critical appraisal tool is reported in Supplementary Tables S1 and S2.

For case–control studies, because of the multifactorial etiology of dental caries, we excluded question 4 ("Was the exposure measured in a standard, valid, and reliable way?"); question 5 ("Was the exposure measured in the same way for cases and controls?"); and question 9 ("Was the exposure period of interest long enough to be meaningful?"). For the same reason, we did not consider question 3 ("Was the exposure measured in a proper and reliable way?") for the critical appraisal of cross-sectional studies.

An assessment of the risk of bias was performed by two authors. Disagreements were solved by a consultation with a third author.

### 3. Results

A total of 6940 papers were considered eligible for title and/or abstract screening, after removing duplicates. After further application of inclusion and exclusion criteria, one hundred and fifty articles were qualified for full-text evaluation. Finally, 16 papers were included in the review: 15 were case–control studies, and 1 was a cross-sectional study (Table 2). Figure 1 depicts the PRISMA flowchart of the selection procedure and the reasons for exclusion. The complete list of the excluded papers after full-text reading is reported in Supplementary Table S3.

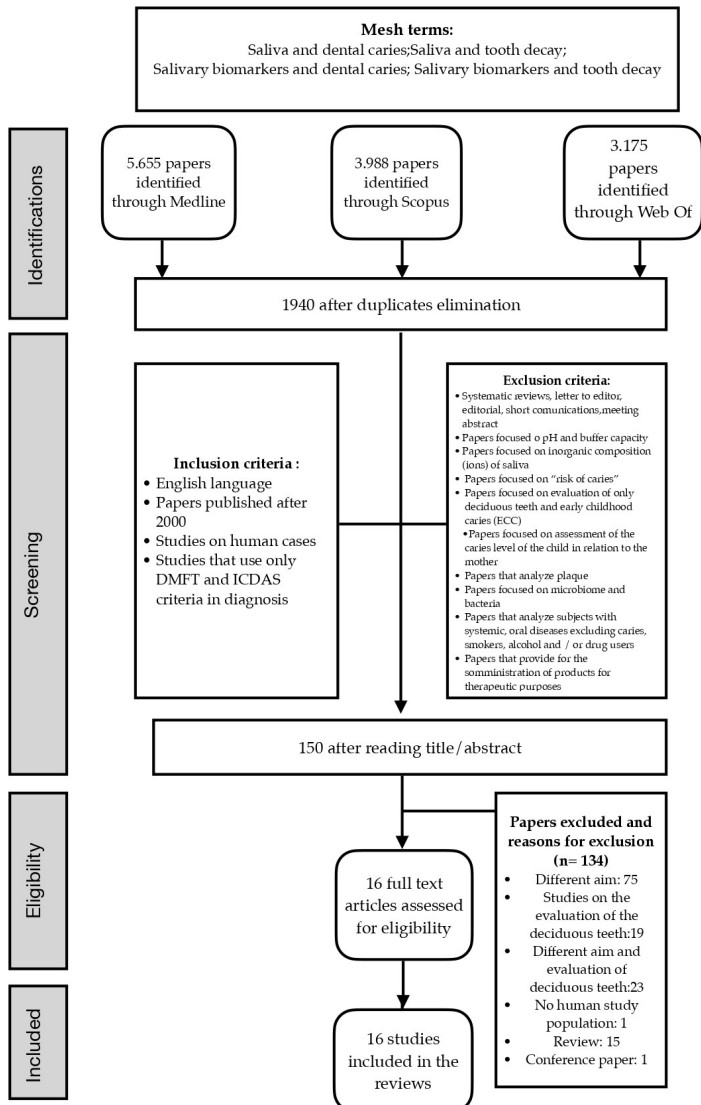

**Figure 1.** Flowchart diagram for the selection of 16 papers included in the review.

*3.1. Study Population (Oral Status; General Health; Diagnosis of Caries)*

Eleven papers [13,15,18–20,22–27] reported data on the oral health of the studied population, detailing that all the subjects exhibited good oral health status (e.g., healthy periodontium and oral mucosa, good oral hygiene, the absence of dental plaque accumulation). Prester et al. verified the absence of burning mouth syndrome and dry mouth condition [26], and only three studies reported on the correlation between saliva flow rates and the presence of caries [14,16,26].

Almost all the papers, except for three [14,18,20], reported data on general health status (e.g., the presence and/or absence of systemic diseases such as diabetes, hypertension, autoimmune disease, medications, diet history).

All the papers, except one [14], reported that the diagnosis of tooth decay was performed by an expert operator with a dental explorer. Only in one paper was an additional radiographic evaluation including digital bitewings and panoramic radiography used [14].

In 15 studies, a standardized criterion for diagnosis was adopted (DMFT in 14 studies and ICDAS in 1 study). The remaining study, considering "open caries lesions", did not mention any standard diagnostic criterion [24]. Eleven studies considered a DMFT score $\geq 3$ to be indicative of a high risk of caries: two studies considered a DMFT score $\geq 11$ [18,20], two a DMFT score $\geq 10$ [16,17], one a DMFT score in the range of 7–12 [21], one a DMFT score $\geq 5$ [13], two a DMFT score $\geq 4$ [27,28], and one a DMFT

score ≥ 3 [22]. Two studies reported average values of DMFT of 38.4 and 11.33 [15,19], and the authors of two papers analyzed salivary biomarkers based on different DMFT score ranges (≤2 and ≥3; 1–3, 4–10, and ≥10) [14,23]. One study included patients with a high DMFT score and analyzed biomarkers based on the presence of dental cavities (a DMFT score of 13.8 with 0 cavities VS a DMFT score of 16.5 with 6.8 cavities—average values) [26]. Only one study considered a DMFT score ≥ 0, with an average value of 2.54 [25]. On the other hand, nine studies considered DMFT = 0 to be indicative of a low risk of caries [14,15,19,22,23,25,27,28,30], one study considered a DMFT score between 0 and 1 as a low risk of caries [21], three a DMFT score ≤ 3 [17,18,20], and only one a DMFT score < 4 [16].

### 3.2. Saliva Collection and Processing Method

Details on the methods for saliva collection and analysis are summarized in Table 3.

Thirteen studies (81%) reported the specific instructions given to patients before saliva collection. The instructions consisted of refraining from eating, drinking (beverages other than water), smoking, and/or practicing oral hygiene for 1 or 2 h before saliva collection. Paqué et al. [24] extended this period to the night before the procedure and Reyes et al. [27] to the previous 12 h.

Eleven articles (69%) reported information about the time of collection. Saliva was collected in the morning (8–12 a.m.), except for one study (2–4 p.m., in addition to 8–12 a.m.) [26].

Piekoszewska-Ziętek et al. reported the collection of salivary fluid using Salivette® collection tubes (Sarstedt AG&Co., Numbrecht, Germany) [25].

The identification of potential salivary biomarkers for the diagnosis of dental caries was conducted on WS samples in all the studies but one [15]. Ayad et al. [15] performed a gustatory-stimulated saliva collection from the parotid glands, using a modified Lashley cup [31].

Specifically, 12 studies (75%) reported the use of passive drooling or unstimulated spitting [13,16,18–25,27,28]; 2 studies (12.5%) evaluated both stimulated and unstimulated WS [17,26]; and 1 study analyzed stimulated saliva [14]. To stimulate salivation, participants were asked to chew pure paraffin wax for 5 min [14,26]. In one study, salivary secretion was stimulated by food ingestion, and three saliva samples were taken at 5, 30, and 60 min after the meal [17].

All the selected studies reported information about the handling and/or storage of saliva. The centrifugation conditions were heterogenous, and the most common parameters were 10,000× *g*, 15 min, and 4 °C. After centrifugation, refrigerated storage was adopted, with temperatures ranging from −20 °C to −80 °C.

Biochemical–analytical methods for the identification and quantification of salivary biomarkers included the following: (1) enzyme-linked immunosorbent assay (ELISA) [14,18–20,22,24,26]; (2) protein electrophoresis [16,23]; (3) real-time polymerase chain reaction (rt-PCR) [25]; (4) high-performance liquid chromatography (HPLC) [15]; (5) liquid chromatography–mass spectrometry (LC-MS) [21]; (6) spectrophotometry [27,28]; (7) enzymatic methods [17,28]; and (8) sialo-chemical analysis [13].

### 3.3. Critical Appraisal

Critical appraisals are summarized in Figure 2A,B for case–control and cross-sectional studies, respectively.

In total, 4 out of the 15 case–control studies received seven "yes" answers to seven questions [15,16,25,26]. Two received six "yes" answers [18,24], seven received five "yes" answers [13,17,20–23,28], and two received four "yes" answers [14,19].

Since all 15 case–control studies reported the same specific diagnostic method for all the subjects of the study, as well as an appropriate statistical analysis, they were positively evaluated for questions 3 and 10.

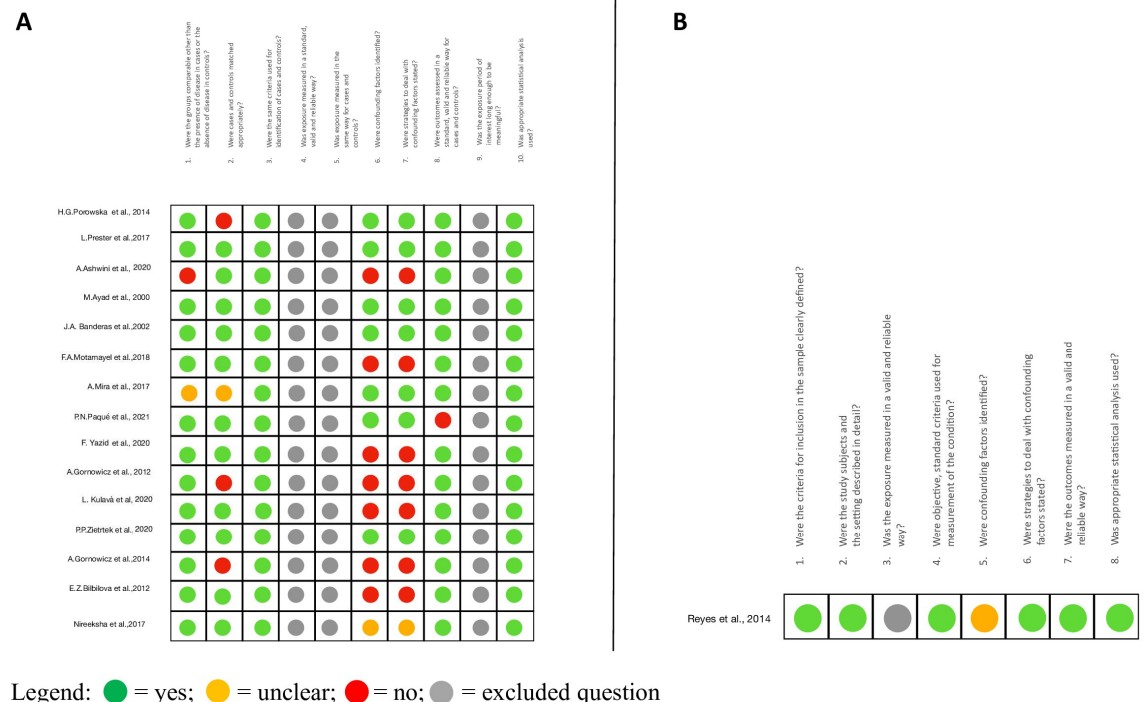

Legend: ● = yes; ● = unclear; ● = no; ● = excluded question

**Figure 2.** (**A**) Critical Appraisal Checklist of the Joanna Briggs Institute (JBI) based on the specific questionnaire for case–control studies [13–19,21–26,28]. (**B**) Critical Appraisal Checklist of the Joanna Briggs Institute (JBI) based on the specific questionnaire for analytical cross-sectional studies [27].

All the papers but two [14,22], received a "yes" answer to the first question. For question 2, the studies by Porowska et al. and Gornowicz et al. [18,20] were negatively evaluated, based on a discrepancy in sample size. The study of Mira et al. [22] received an "unclear" answer for question 2 due to the limited information relating to the two study groups (cases and controls).

Eight papers [13,14,17,19–21,23,28] received "No" or "Unclear" answers to questions 6 and 7 as they did not mention the potential confounding factors and/or their normalization by means of the statistical analysis.

One paper did not satisfy question 8 [24] because of the absence of a standard and reliable method to identify caries cases (e.g., no mention of the DMFT or ICDAS criteria).

The only cross-sectional study [27] received six "yes" answers to seven questions. The authors did not report on confounding factors ("Unclear" answer to question 6). However, they used the Shapiro–Wilk and Levene tests to determine the normality of data distribution and variance homogeneity (a "yes" answer to question number 6).

With ratings ranging from 4 to 7 on the JBI Critical Appraisal Checklists, the 16 included papers were considered to be of moderate to high quality.

Only one disagreement was raised during the evaluation step for the paper of Ashwini et al. 2020, which was solved by the third reviewer's appraisal.

### 3.4. Salivary Biomarkers

Overall, 12 papers (75%) analyzed only protein biomarkers [14–16,18–21,23,24,26–28], 2 papers (12.5%) only metabolite biomarkers [13,17], 1 papers (6.25%) both protein and gene biomarkers [25], and 1 paper (6.25%) both protein and metabolite biomarkers [22].

Details on the biomarkers evaluated in each study are summarized in Table 4.

Salivary molecules that are significantly associated with the presence of dental caries were classified as proteins, genes, or metabolites. These molecules are potential biomarkers for dental caries diagnosis.

### 3.4.1. Protein Biomarkers

Mucins

Nireeksha et al. found that salivary mucin levels were increased in caries-active subjects ($p < 0.05$), but the authors did not report details on specific mucins [18,23]. Instead, Gabryel-Porowska et al. analyzed the concentration of three salivary mucins: mucin-1 (MUC1), mucin-5B (MUC5B), and mucin-7 (MUC7) [18]. Significantly higher MUC1 levels were found in subjects with DMF > 11 when compared to subjects with DMF = 3 ($p = 0.011$), confirming the correlation between MUC1 and the presence of caries. MUC5B and MUC7 salivary concentrations were not significantly associated with this dental pathology ($p = 0.06$ and $p = 0.918$, respectively). In another work, Banderas-Tarabay and colleagues analyzed a series of proteins [16]. Interestingly, subjects with a higher DMFT index (=11.87) showed a significant reduction or the absence of high-molecular-weight mucin glycoprotein-1 (MG1, encoded by MUC5B) and low-molecular-weight mucin glycoprotein-2 (MG2, the translational product of MUC7), ($p \leq 0.001$). Salivary alpha-amylase levels were not statistically associated with caries, whereas subjects with a higher DMFT index (=10.0) presented lower levels of acidic proline-rich protein-1 (PRAP-1) ($p \leq 0.001$).

Glycoproteins, Immunoglobulins, and Enzymes

Prester et al. investigated the role of soluble CD14 (sCD14) in the unstimulated and stimulated saliva of patients with dental caries. Median levels of sCD14 were higher in the active caries than in the caries-free group in both stimulated and unstimulated saliva ($p < 0.01$) [26].

Gornowicz et al. proved that patients with high dental caries activity (DMF > 11) had significantly increased levels of secretory IgA, histatin-5, and lactoperoxidase (LPO) compared to subjects with lower caries activity ($p < 0.05$) [20] Conversely, Nireeksha et al. reported that salivary IgA (sIgA) levels in subjects with active caries were decreased with respect to caries-free subjects ($p < 0.05$) [23].

It is worth detailing the results of the study of Kulhavá et al. [21]. They analyzed the supernatant and pellet fractions of salivary samples obtained from subjects with and without caries. Fourteen proteins showed higher expression levels in the supernatant samples of caries-free subjects compared with subjects with dental caries.

Three proteins (annexin A1, cornulin, and 14-3-3 protein ς) had higher expression in pellet samples than in the supernatants of subjects with caries ($p < 0.5$).

Alpha-amylase was also investigated by Banderas-Tarabay et al. and Yazid et al. [16,28]. Even if Banderas-Tarabay and colleagues did not attribute diagnostic relevance to alpha-amylase, Yazid et al. observed a significant increase in the alpha-amylase absorbance signal (UV-Vis spectroscopy) in patients with caries

In the study performed by Piekoszewska-Ziętek et al., salivary carbonic anhydrase isozyme VI (CA VI) levels were significantly lower in patients with dental caries ($p = 0.014$) [25].

Reyes et al. investigated the role of urease and arginine deiminase system (ADS) activity in saliva and supragingival plaque. Urease activity was significantly higher both in the saliva (3.024 vs 0.437 $\mu$mol min$^{-1}$ mg prot.$^{-1}$, $p = 0.010$) and plaque (18.120 vs 0.370, $p = 0.033$) of caries-free subjects. Also, ADS activity in saliva (6.050 vs 1.350, $p = 0.0154$) and plaque (8.830 vs 1.210 $\mu$mol min$^{-1}$mg prot.$^{-1}$, $p = 0.025$) was higher in individuals with DMFT = 0 compared to caries-active patients, having at least four teeth with active caries [27].

Interleukins and Chemokines

Paqué et al. evaluated 19 cytokines, seven chemokines, four growth factors, two metalloproteinases, one metallopeptidase inhibitor, one protease, and the presence of 10 oral bacteria (*P. gingivalis, T. forsythia, T. denticola, F. nucleatum, C. rectus, P. intermedia, A. actinomycetemcomitans, S. mutans, S. sobrinus, and oral lactobacilli*) in healthy individuals and patients with gingivitis or caries. Significantly higher levels were observed for interleukin 13 (IL-13) ($p = 1.5 \times 10^{-15}$ caries/gingivitis, $p = 4.0 \times 10^{-13}$ caries/healthy), interleukin

2-RA (IL-2-RA) ($p = 3.3 \times 10^{-6}$ caries/gingivitis, $p = 1 \times 10^{-4}$ caries/healthy), interleukin 4 (IL-4) ($p = 1.5 \times 10^{-15}$ caries/gingivitis, $p = 4.1 \times 10^{-13}$ caries/healthy), and Eotaxin/CCL11 ($p = 8.1 \times 10^{-5}$ caries/gingivitis, $p = 4.4 \times 10^{-4}$ caries/healthy) in patients with caries when compared to the other groups (healthy and gingivitis) [24].

Gornowicz et al. found a statistically significant increase in IL-6, IL-8, and TNF-alpha levels in the unstimulated WS of subjects with dental caries compared to the controls ($p < 0.05$) [19].

### Peptides

Ayad et al. analyzed the phenotypes of 18 genes (proline-rich protein phenotypes—*Pe, Pmf, Ps1, Con2, PmS, Con1, G11, G12, G13, G14, Po, Db, Pa, Pif, PR1, PR2, Pmo1, and Pc2*), which were tested for differences between caries-free and caries-susceptible subjects. *Ps1* and *Con1* peptides were more common in the caries-free group than in the caries-susceptible group ($p = 0.046$). The difference in prevalence for *Pmo 1* was close to statistical significance, thus suggesting that its prevalence might be lower in the caries-free group ($p = 0.06$) [15].

### Other Proteins

Ashwini et al. investigated matrix metallopeptidase 8 (MMP-8) and discovered a significantly higher concentration of this molecule in patients with caries compared to the controls ($p < 0.05$) [14].

To test the hypothesis that some molecules were mainly produced after dietary carbohydrate fermentation, Mira et al. compared 25 salivary compounds in caries-free and caries-active individuals at different time points of dental biofilm formation and times of the day. Based on the *p*-values ($p < 0.5$), the following salivary proteins were proven to discriminate between healthy and caries-active individuals: LL-37, IgA, statherin, and fibronectin (statherin only in saliva collected after a sugary solution rinse) at 30 min after toothbrushing (morning sample), and β-defensin 2, LL-37, collagen I, and fibronectin at 6 h after toothbrushing (afternoon sample) [22].

### 3.4.2. Genes

Piekoszewska-Ziętek et al. examined three single nucleotide polymorphisms (SNPs) of the carbonic anhydrase (CA) VI gene (*rs2274327*; *rs2274328*; *rs2274333*) in buccal smear. No association between the increased or decreased risk of caries and the analyzed polymorphisms was found. However, some significant positive correlations were found between the *rs2274333* A/G genotype and the presence of active white spot lesions ($p < 0.05$). Moreover, there were some significant relations concerning SNPs and the salivary buffer capacity and flow rate in *rs2274327* and *rs2274328* [13,25].

### 3.4.3. Metabolites

Ahmadi-Motamayela and coworkers investigated the salivary and serum malondialdehyde (MDA) levels. Their results showed significantly higher levels of MDA in the case group compared to the healthy control group ($p = 0.001$).

Zabokova Bilbilova et al. examined the values of salivary urea in subjects with different caries activities. The salivary concentration of urea was significantly lower in patients with a high DMFT index (DMFT > 10, from 3.4 to 5.5 mmol/L) compared to subjects with low caries index (DMFT= 0–3, from 5.5 to 9.1 mmol/L). The same result was obtained with the concentrations of salivary urea measured at 5, 30, and 60 min after the meal [17].

Mira et al. found that formate and phosphate at 6 h after toothbrushing (afternoon sample) and phosphate and lactate at 30 min after toothbrushing (morning sample) were able to distinguish healthy subjects from caries-active individuals [22].

## 4. Discussion

The present systematic review aimed to emphasized the relevance of groups of salivary molecules that are significantly associated with the presence of dental caries by comparing individuals with and without caries experience.

Dental caries represents a complex disease that, if diagnosed and treated early, can be stabilized and, in some cases, reversed with the remineralization of the tooth surface [32].

Caries diagnosis is usually performed during the dental visit by a general dentist [33]. The traditional caries detection method includes a careful visual inspection, dental probing, and radiographic examination if needed. In most cases, this methodology is reliable for detecting caries that have progressed into dentine and need conservative treatments [34]. However, early-stage caries (e.g., those producing small changes in dental enamel) are challenging to diagnose visually and radiographically [6]. Innovative methods for the early diagnosis of carious lesions are currently available [7]. Some of such new approaches could lead to a decrease in invasive treatments and costs for patients and health care systems.

Dental caries is in direct contact with saliva, and some of its components react to the acidic environment induced by bacterial metabolism, contrasting their biofilm's development and adhesion. Indeed, the detection and measurement of salivary caries biomarkers may represent an attractive alternative for the early diagnosis of caries. However, using a single biomarker predictive of disease occurrence appears unsuitable given the multifactorial etiology of caries.

According to the results presented, the most investigated molecules are alpha-amylase [16,21,28] and mucins [16,18,23]).

Salivary alpha-amylase is involved in maintaining oral homeostasis. The alteration of this enzyme is associated with dental caries development, leading to a dysregulation of enamel calcium-binding mechanisms and modifying the capacity of this enzyme to bind to oral streptococci [35]. Even if Banderas-Tarabay et al. [16,21,28] did not attribute diagnostic relevance to alpha-amylase, Yazid et al. and Kulhava et al. highlighted a significant association of the alpha-amylase levels with the presence of dental caries [21,28]. However, these studies showed contrasting results.

The increase in the alpha-amylase absorbance signal in patients with caries is justified by the binding of the enzyme to oral microorganisms, which facilitates starch hydrolysis inside the biofilm and the acid production mechanism of dental plaque, causing dental caries [21,28]. Kulhava and collogues found that salivary alpha-amylase had significantly higher expression levels in the supernatant samples of caries-free subjects [21]. This result could confirm the hypothesis that the binding of alpha-amylase to bacteria in solution may be considered protective if it leads to bacterial clearance from the oral cavity [35]. To date, it is unclear which of these alternate hypotheses is correct.

The reason for such a discrepancy may partly lie in the different biochemical approaches (UV-Vis spectroscopy and LC-MS, respectively) used to evaluate salivary samples.

Mucins are proteins correlated with the formation and progression of dental caries. These molecules constitute an important class of salivary glycoproteins. Notably, they account for approximately 20–30% of the total proteins in unstimulated saliva [36] and play a variety of functions critical to maintaining a stable oral defense. As part of the enamel pellicle, mucins favor the colonization of certain microorganisms while promoting the clearance of others, thus contributing to the formation of a selective/protective barrier; their hydrophilic properties protect oral tissue surfaces against mechanical wear; and they prevent acids access, thus limiting mineral erosion from tooth surfaces [37]. The results of the studies of Gabryel-Porowska et al. and Nireeksha et al. showed a correlation between mucin 1 levels (MUC1) and the DMFT index [18,23]. The increase in mucin levels during caries development could represent a protective mechanism that counteracts acid and bacterial impacts. However, Gabryel-Porowska et al. showed that mucins were not significantly enhanced in cases of an extremely high DMFT index [18]. On the other hand, the results of Banderas-Tarabay et al. showed that a decrease in the salivary level of mucins was associated with a higher DMFT index, a symptom of a serious oral health decline [16].

Based on this incongruity, it can be speculated that different mucin levels might correspond to different stages of caries development.

The results obtained with sIgA salivary levels are also controversial. Gornowicz et al. suggested that a massive presence of caries can be associated with high levels of sIgA, probably increasingly secreted to potentiate their antibacterial effect [20]. Such results are supported by similar studies not included in the present review [38–40]. On the other hand, Nireeksha et al. found that the sIgA level was decreased in active-caries patients [23]. The authors attribute this finding to sIgA's highly specific binding ability to microbial species, resulting in bacterial inactivation and the prevention of adhesion. Some studies have reported a correlation between sIgA levels and the subjects' age. Jafarzadeh et al. demonstrated that mean salivary sIgA levels increased with ages up to 60 years and then slightly decreased in subjects aged 61–70 years [41]. However, the age range of the subjects involved in these two reviewed studies was different (18 yrs in Gornowicz et al. and 25–40 yrs in Nireeksha et al.). It might be relevant to carry out further research correlating the level of salivary sIgA in patients with caries in the same age range.

Among the molecules that have been described singularly, there are proteins with buffering capacity [25] and antimicrobial proteins with enzymatic activity, such as lactoperoxidase (LPO) and lysozyme C. Kulhava et al. reported a significant up-regulation of lysozyme C and other immune proteins dissolved in the salivary supernatants of caries-free subjects and suggested that they could play an important role in caries prevention [21,42]. The authors also analyzed the corresponding salivary pellets and revealed a higher concentration of three calcium-binding proteins (annexin A1, cornulin, and protein ς) in caries-positive subjects. This finding might relate to the decalcification of enamel during the carious process. The concentration of annexin A1 was also significantly higher in the caries-free supernatant. These results appear to be contrasting but might reflect the various and complex roles of annexin A1 (e.g., in innate immune response as an effector of glucocorticoid-mediated responses, a regulator of the inflammatory process, its high affinity for $Ca^{2+}$).

Urea is part of the saliva buffering system that can neutralize the oral cavity's acids [42]. As reported by Zabokova Bilbilova et al., the salivary level of urea was significantly lower in patients with higher DMFT indexes [17,27]. This result agrees with the study of Reyes et al. [27], which demonstrated higher urease levels in individuals with low DMFT scores. Accordingly, caries-free subjects produced higher ammonia levels because of the salivary urease and arginine deiminase systems. Moreover, increased production of alkaline substances was associated with a low incidence of dental caries, suggesting that they might be investigated as predictive salivary biomarkers for dental caries [27].

One limitation of the present systematic review is that the authors employ different DMFT values to establish a high risk of caries. Despite the fact that the majority of studies (11 out of 16) utilize a DMFT score $\geq$ 3, there is currently insufficient scientific evidence to consider this as a threshold value.

Due to the complex interactions between salivary components and functions, it may thus be unrealistic to expect any single salivary factor to effectively identify caries-susceptible individuals. In fact, most of the molecules analyzed are found in saliva following the distinct mechanisms of production and release in the oral fluid. It is likely that a pool of salivary biomarkers should be assessed in conjunction with other caries risk factors and indicators, such as diet, exposure to fluoride, individual salivary flow rate, and sociodemographic and behavioral variables, in order to determine caries risk [43].

It is noteworthy that this review highlighted great homogeneity regarding the selection of the study population and the saliva sample collection procedure. All studies included subjects with a good oral and general health status and, except one [15], used whole saliva, centrifuged and fast-frozen at temperatures between $-20\,^{\circ}C$ and $-80\,^{\circ}C$.

All the included studies, except for two [15,16], were published within the last 10 years, highlighting the recent interest in salivary biomarkers related to dental caries. The critical appraisal of the present systematic review attributed a "moderate" (7 out of 16 studies,

44%) or "good" (9 out of 16 studies, 56%) quality level to the included studies. Furthermore, all case–control studies appropriately matched study groups, using the same identification criteria, and used appropriate statistical analyses to correlate salivary biomarkers with dental caries. These features markedly decrease the risk of bias and support the high quality of all the studies included in this systematic review.

Differently from other systematic reviews, we focused only on permanent dentition and excluded studies on dental caries in children, as this may be due to a different pathogenetic mechanism. Furthermore, most of the published systematic reviews deal with a single category of molecules or a limited number of microorganisms. Indeed, according to the present paper, only a small section of the current literature focuses on molecules other than proteins. A review published in 2022 confirms the important role of some proteins also studied in the present research (e.g., alpha-amylase, histatin-5, lactoperoxidase, and carbonic anhydrase VI) [30]. This review instead considers the entire array of salivary components, including proteins, metabolites, and genes, that have been identified to date [30]. Based on our study selection, some protein levels are more likely to be involved in the occurrence of dental caries with respect to other groups of molecules, albeit the selected studies occasionally gave contradictory results. This occurrence highlights the necessity for further research as well as the development of appropriate and comparable experimental settings and conditions.

Other categories of potential biomarkers have yet to be identified and thoroughly investigated. Future developments in salivary metabolomics, genomics, and transcriptomics may give additional impetus to this research.

## 5. Conclusions

Most of the salivary molecules presented in this review might potentially play an important diagnostic or predictive role. According to the "good" quality studies, salivary mucins, glycoproteins (sCD14), interleukins (IL-2RA, 4,-13), urease, carbonic anhydrase VI, and urea appear to exhibit significant different levels in healthy and active-caries subjects [15,17,18,24–27]. These salivary molecules should be the target of clinical research to validate or exclude their relevance as biomarkers for dental caries.

Acknowledging its non-invasiveness and ubiquitous applications, saliva as a probing biofluid sample remains highly attractive. Indeed, salivary diagnostic/prognostic tools are less invasive and less harmful than current tools and enable physicians to intervene early, possibly altering the course of the disease and significantly reducing suffering and disability in patients. In dental practice, a salivary test may be useful to assess the presence of dental caries when it is challenging to perform an X-ray validation (e.g., pregnant woman, patient with disability) or when X-ray is not predictable (e.g., interproximal caries with dental overlap).

From a future perspective, the early self-diagnosis of dental caries might be accomplished through salivary tests potentially available for those categories of patients considered to be at a high risk of caries.

**Supplementary Materials:** The following supporting information can be downloaded at: https://www.mdpi.com/article/10.3390/cimb46050258/s1. Table S1. "JBJ Critical appraisal Checklist" for case control studies. Table S2. "JBJ Critical appraisal Checklist" for cross sectional studies. Table S3. Papers excluded and the reasons for exclusion.

**Author Contributions:** R.A., V.M. and M.M. contributed to the conceptualization, design, and data interpretation and drafted and critically revised the manuscript. T.A.P., E.F. and M.G. contributed in designing the work and to the interpretation of data and critically revised the manuscript. S.P. and P.V. contributed to critically revising the manuscript. All the authors have confirmed the final approval of the version to be published and agree to be accountable for all aspects of the work in ensuring that questions related to the accuracy or integrity of any part of the work are appropriately investigated and resolved. All authors have read and agreed to the published version of the manuscript.

**Funding:** This research received no external funding.

**Informed Consent Statement:** Not applicable.

**Data Availability Statement:** All data generated or analyzed during this study are included in this article. Further enquiries can be directed to the corresponding author.

**Conflicts of Interest:** The authors declare no conflicts of interest.

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
