# Peer review of "Salivary Diagnosis of Dental Caries: A Systematic Review"

_cimb, doi:10.3390/cimb46050258_

Round 1

Reviewer 1 Report

Comments and Suggestions for Authors

This article is dealing with an interesting topic that could has potential application in clinical practice and patient education as well.

Summary is well structured and well informational.

Introduction has covered all the important findings of this topics up to this date.

Materials and methods are done on the todays principles and protocols for systematic review.

Discussion provides all major results with comparison with other significant researches in this field. Authors addressed unanswered salivary biomarkers, leaving new ideas for the future investigations. 

Conclusion answered to the proposed question in the objectives of this article. 

Altogether, this is very well structured and written review article, well done. 

Reviewer 2 Report

Comments and Suggestions for Authors

Dear Authors,

I have completed my evaluation of the article. I appreciate the authors’ efforts in preparing this manuscript. These are my comments:

1) Please sort key words alphabetically.

2) Abstract:

A.  Please provide more details about some salivary compounds in the conclusion. 

3) Introduction:

A.  Please expand this paragraph, and this reference [https://pubmed.ncbi.nlm.nih.gov/32869220/ DOI: 10.1055/s-0040-1715708] can be helpful.

Saliva is a complex biological fluid produced by parotid (20%), submandibular (65%-68 70%), sublingual (7% to 8%), and minor (<10%) salivary glands, as well as by gingival 69 sulcus (crevicular fluid) [8]. Whole saliva (WS) consists of a pool of different oral fluids. 70 Besides the massive presence of water, saliva contains metabolites, enzymes, antibodies, 71 hormones, antimicrobial molecules, and cytokines, all potential biomarkers for oral and 72 systemic diseases [9].

4) Materials and Methods:

A: Please delete the brackets and revise it as follows: Two independent researchers performed a first-level screening by evaluating titles and abstracts.

B:  All the tables and figures are informative. 

Critical appraisal

A: Please elaborate on all questions (i.e., 1–9) in this paper.

B: Please revise as follows: Assessment of the risk of bias was performed by two authors. Disagreements were solved by a consultation with a third author.

5) Results:

A: Based on the results of 16 studies included in the present study, some salivary compounds play a crucial role in the diagnosis of caries.

6) Discussion:  

A: This part is well written.

7) Conclusion:

A: This part is well written.

Reviewer 3 Report

Comments and Suggestions for Authors

This paper reviewed the involvement of salivary constituents in the diagnosis of dental caries. The authors concluded that some salivary compounds might be involved in the development/presence of dental caries. The authors reviewed a very short period of time (June 2000 to October 2013), why was such a restricted period of time chosen. Furthermore, human studies with detailed information on dental caries AND specifically applying DMFT and/or ICDAS criteria were included. Why the DMFT/ICDAS criteria? Not directly related to salivary constituents. When reading the studies included, a wide variation of biomarkers was assessed, but there was no common mechanism involved. Furthermore, only caries versus non carious was assessed and not subject prone to developing dental caries. Thus, in patients with caries, the saliva might be on the basis but also a result of dental caries. Furthermore, the flow rates should be included (and related oral flora) as this might be more on the basis of dental caries than the composition of saliva itself.

Reviewer 4 Report

Comments and Suggestions for Authors

There have been multiple systematic reviews on this same subject. Please outline what is novel in this review to warrant consideration for publication.

In the abstract – the p value was only stated as 0.5. This is not statistically significant.

The search strategy is not adequate, especially given that the authors mentioned they abided by PRISMA guidelines. Please refer to https://www.bmj.com/content/372/bmj.n160 for the 2020 version.

Please explain why the search was only conducted from June 2000 onwards.

The format of the review is odd. Table 2-4 comes under Results section. The methodology of the data collection process, data items, effect measures, data synthesis, causes of heterogeneity should be clearly stated here. The authors are not abiding by PRISMA guidelines.

Please provide a reference/reason for utilising JBI for the present review.

The authors mentioned 16 studies were included in final review. However, on the PROSPERO website, it stated 19. Please explain this discrepancy.

The authors mentioned that 150 articles were selected for full-text review. 16 studies were selected. It is also useful to list studies that were potentially relevant but for which the full text or data essential to inform eligibility were not accessible. This information can be reported in the text or as a list/table in the report or in an online supplement. Potentially contentious exclusions should be clearly stated in the report. (See Item 16B of PRISMA). This brings to mind the previous point on why there is a discrepancy between this review and the PROSPERO protocol on the website.

There was not enough information on caries diagnosis (extent, number of teeth, sites etc.). in section 3.1. Table 4 also does not contain sufficient information on the association between saliva biomarkers and caries – the authors have treated dental caries as presence/absence, when this should be more clearly refined (DMFT above a threshold value, for example). In fact, this should be clearly highlighted in the manuscript – are the authors trying to investigate the use of salivary biomarkers in diagnosing presence/absence of caries, or threshold levels of caries in the individual. Saliva diagnostics is patient-level, and I find it hard to believe that it will be sensitive to diagnosing a single cavity within the mouth – and the included studies do not suggest that either. The manuscript needs to make clear this clear from the outset (in the abstract, intro, discussion), and should also highlight this in the discussion. There was no mention of this (patient-level vs site-level issues) within the manuscript.

Lines 281-288 seems out of place in a review.

Round 2

Reviewer 3 Report

Comments and Suggestions for Authors

I still consider DMFT not as proper index for the purpose of this study (to find salivary biomarkers of caries development). DMFT is an indirect measure of caries better to be used in epidemiological studies. Not for mechanistical studies. Although you mentioned this well in your study that this is a major limitation, I am sorry that I cannot be  in favour of your study.

Reviewer 4 Report

Comments and Suggestions for Authors

Well done! The paper is ready for publication and I congratulate the authors.